# Distinguishing Genetic Drift from Selection in Papillomavirus Evolution

**DOI:** 10.3390/v15081631

**Published:** 2023-07-26

**Authors:** Robert D. Burk, Lisa Mirabello, Robert DeSalle

**Affiliations:** 1Departments of Pediatrics, Microbiology & Immunology, Epidemiology & Population Health, Obstetrics, Gynecology and Woman’s Health, and Albert Einstein Cancer Center, Albert Einstein College of Medicine, Bronx, NY 10461, USA; 2Division of Cancer Epidemiology and Genetics, National Cancer Institute, National Institutes of Health, Rockville, MD 20850, USA; 3Sackler Institute of Comparative Genomics, American Museum of Natural History, New York, NY 10024, USA

**Keywords:** molecular evolution, genetic drift, human papillomavirus (HPV)

## Abstract

Pervasive purifying selection on non-synonymous substitutions is a hallmark of papillomavirus genome history, but the role of selection on and the drift of non-coding DNA motifs on HPV diversification is poorly understood. In this study, more than a thousand complete genomes representing *Alphapapillomavirus* types, lineages, and SNP variants were examined phylogenetically and interrogated for the number and position of non-coding DNA sequence motifs using Principal Components Analyses, Ancestral State Reconstructions, and Phylogenetic Independent Contrasts. For anciently diverged *Alphapapillomavirus* types, composition of the four nucleotides (A, C, G, T), codon usage, trimer usage, and 13 established non-coding DNA sequence motifs revealed phylogenetic clusters consistent with genetic drift. Ancestral state reconstruction and Phylogenetic Independent Contrasts revealed ancient genome alterations, particularly for the CpG and APOBEC3 motifs. Each evolutionary analytical method we performed supports the unanticipated conclusion that genetic drift and different evolutionary drivers have structured *Alphapapillomavirus* genomes in distinct ways during successive epochs, even extending to differences in more recently formed variant lineages.

## 1. Introduction

Hundreds of different papillomaviruses have been described [1] encompassing the full range of vertebrate hosts from fish [2] and amphibians [3] to birds [4] and especially mammalian host groups [1]. Several dozen species and more than 200 types have been curated from humans [5,6,7,8]. *Alphapapillomavirus* genomes have been the most scrutinized in light of their role in cervical and other cancers [9,10,11]. The *Alphapapillomavirus 9* species group of human papillomavirus (HPV) types, and HPV16 in particular, stand out as being most strongly associated with carcinogenesis [8,12,13,14,15,16]. Thus, understanding how their evolution has resulted in these devastating human pathogens [17,18] is important.

Considerable effort has been devoted to assessing whether there are amino acid differences that would readily explain the carcinogenic properties of HPV16, the species *Alphapapillomavirus 9*, and the more encompassing high-risk (HR) clade of *Alphapapillomavirus* [19,20,21,22]. Evolutionary selection is typically measured on the basis of rates of nonsynonymous to synonymous substitutions in codons [23,24,25]. Molecular evolutionary biologists have few other tools to convincingly discover and describe evidence of evolutionary pressures acting at more fundamental genome structural or basic nucleotide composition levels. Thus, current evidence for selection or its alternative, genetic drift, are hampered by few analytical methods. Whole genome analyses of various HPV types do not support the notion that pathogenicity is more strongly connected to nonsynonymous substitutions [26] than, perhaps, to a lack thereof [27,28,29] or to other kinds of substitution in papillomavirus genomes [29,30]. Indeed, elevated oncogenic risk from large epidemiological studies is also associated with variations in non-coding regions and with silent substitutions that are non-randomly distributed across the ~8000 bp genome [30,31,32,33]. Comparative phylogenetic perspectives on papillomavirus genome evolution may yet reveal how disease phenotypes correlate with higher-level clade membership, with species membership, with viral type, and even with lineage and sublineage distinctions [31,34,35,36,37].

Chen et al. [38] generated UPGMA trees based on k-mer spectra encompassing all papillomaviruses that corresponded surprisingly well with classical alignment-based phylogenetic trees. Those results indicated that changes in amino acid sequences are not the only consequence of evolutionary pressure. Beyond codons, the depletion of CpG sites appeared to be phylogenetically structured [38], which is relevant given the association of CpG content with nucleosome formation and for methylation and deamination [31,39,40,41,42,43,44,45,46,47]. Similarly, host class 3 apolipoprotein B mRNA-editing enzyme (APOBEC3) anti-viral activity should select against TCA and TCT sites [27,28,48,49,50,51]. However, the early open reading frames E6 and E7 upregulate APOBEC3 and host methyltranserferase activity [49,52,53,54,55], but they have strikingly different patterns of mutability. Oncogenic HPV16 non-synonymous single nucleotide polymorphism (SNP) variants are hypovariable in E7 relative to other open reading frames [28]. In contrast, the E6 locus appears to be able to vary more freely than other early- or late-expressed gene sets [28,56].

In this report, we use phylogenic approaches to investigate non-coding genomic motifs that have previously been of interest regarding the structure and evolution of papillomavirus genomes. Non-recombining asexual organisms, such as papillomaviruses, under continual mutational burden display some aspects of selective pressure to avoid host primordial defenses, such as cytosine deamination [27,43,57,58]. Here, we analyzed the intrinsic nucleotide composition (i.e., A, C, G and T) already speculated to be of lineage-specific significance [59,60,61,62], the number and distribution of CpG sites, sites available to APOBEC3 attack, and strand disparities in APOBEC3 sites. We also studied guanine quadruplexes [63], other guanine-rich motifs (e.g., duplexes) [64], toll-like-receptor 9 (TLR9) stimulatory and suppressing sequence motifs implicated in viral pathogenicity [65,66,67,68,69,70], high-affinity and non-canonical E2-binding sites [71,72], inverted repeats, perfect palindromes, duplicated regions, and reverse complementary regions [73,74]. The evidence regarding the evolution of DNA motifs in HPV genomes is most consistent with genetic drift.

## 2. Materials and Methods

### 2.1. Virus Genome Data

For higher level analyses at the genus level, all reference genomes for each of the 83 *Alphapapillomavirus* reference types were obtained in Genbank format from the Papillomavirus Episteme database (PAVE, https://pave.niaid.nih.gov, accessed on 1 April 2023) [5]. For the purposes of phylogenetic and UPGMA analyses, two datasets were established: one unaligned and without modification and a second aligned dataset in which each genome was informatically processed to remove both non-coding and overlapping coding regions on the basis of annotations embedded in the Genbank format. For phylogenetic and ancestral state reconstructions of *Alphapapillomavirus 9*, reference genomes were obtained from PAVE for each variant of *Alphapapillomavirus 9* types, as well as the variants of HPV18 and HPV45 as outgroup taxa. In order to maximize assessment of variability for Principal Components Analysis within *Alphapapillomavirus 9* variants, whole genomes were obtained from NCBI in which no more than 25 nucleotides were missing or ambiguous in the upstream regulatory region (URR) and annotated for all open reading frames. This included a total of 747 genomes for HPV16, 284 for HPV35, 41 for HPV31, 28 for HPV33, 138 for HPV58, and 35 for HPV67. Many deposited complete genomes for HPV52 were found to be incompletely annotated. Relaxing the annotation requirement permitted expansion of the lineage representation for HPV52 to 191 complete genomes.

### 2.2. Phylogenetic and Cluster Analysis

Phylogenetic trees at the *Alphapapillomavirus* species and type levels were obtained with maximum likelihood (GTR+G) using PAUP 4.0a169 [75] analyzing the concatenated nucleotide sequences from non-overlapping open reading frames for E6, E7, E1, E2, L2 and L1 separately aligned according to the aligned inferred amino acid sequences using the TranslatorX server (translator.co.uk, accessed on 1 April 2023) [76]. Because codon models of selection must distinguish between synonymous and nonsynonymous nucleotide substitutions in-frame [77], we excluded all overlapping reading frames from those analyses (for example, but not limited to, where E4 and E2 overlap). In addition to phylogenetic trees, the separate unweighted pair group method with arithmetic mean (UPGMA) trees were constructed based on codon usage, trimer composition, and nucleotide composition. For each of the these, Euclidean distances were constructed using reference types from PAVE [5]. Distance matrixes were evaluated with UPGMA in the *fitch* package of Phylip version 3.695 [78]. Trees were visualized with FigTree version 1.4.3 [79].

### 2.3. Enumeration of Sequence Motifs

Code was written in Python 3 leveraging the Biopython module to determine the number and position of nucleotide sequence motifs across each whole and unaligned genome analyzed, both in total and in a sliding window scan of the genomes. The level of resolution (sliding window size of 150 nucleotides) was chosen to be small enough (i.e., half the size) to resolve the two smallest open reading frames (E7 and E4), while also being large enough to capture degrees of freedom on the occurrence of DNA motifs. For example, to the extent that CpG motifs are suppressed in the *Alphapapillomavirus 9* lineages, even if those motifs were randomly distributed, one would expect to find a CpG motif only once every 75 dimers. Each iteration involved a 9-nucleotide widow shift to have at least 30 shifts in E6 and E4. Parameters recovered with regular expression (regex) pattern matching (see Appendix A) in unaligned genomes as well as base composition on the coding strand (i.e., A, C, T, G), trimer composition on the coding strand, CpG motifs, high affinity E2 binding site motifs on the coding and opposite strand, APOBEC3 binding site motifs on both strands, toll-like receptor 9 (TLR9) stimulatory and suppressing motifs on both strands, guanine quadruplexes on both strands, palindromes, near-palindromic inverted-repeats allowing from 3 to 50 spacing nucleotides, larger and more distant duplicated or duplicated, and, finally, reverse-complemented regions being at least 14 nucleotides long and separated by at least 14 nucleotides. Trimer composition was assessed by way of a normalized Euclidean distance in a sliding window relative to that of the whole genome.

### 2.4. Statistical Analyses

Determinations of confidence limits and statistical significance of local trimer composition, GC content, and the relative number of CpG and APOBEC3 sites in each 150 nt sliding window were assessed by comparison of observed values to those obtained from a same-sized window pseudo-sampled from 1000 randomizations of the entire genome. Assessment of differences in relative proportions of covarying residues and relative proportions of APOBEC sites on opposite strands were accomplished through standard Z score calculations with a Bonferroni correction for multiple comparisons.

Principal Components Analysis (PCA) was accomplished in Python 3 with the SciKit-Learn library, using a standard scaler for preprocessing and Seaborn for visualization.

In order to obtain unbiased branch lengths for each DNA sequence motif and for changes in base composition, ancestral state reconstructions of these continuous variables were fitted to the ML topology using a Brownian motion model in Mesquite version 3.70 [80]. For each motif and branch combination, the number of changes was determined to be significantly different where that exceeded the upper 95% confidence threshold for that motif across all branches. Phylogenetic Independent Contrasts (PIC) of reconstructed ancestral states were calculated in Python 3 with the Denropy library. Pearson product moment correlation *p*-values among continuous variable PIC were converted to false discovery (Q) values through the ratio of their relative rank to the total number of comparisons.

Non-overlapping reading frame alignments were examined using codon models with HyPhy [81] for overall rates of nonsynonymous to synonymous substitution (with BUSTED), site-by-site episodic selection (with MEME), and/or changes in the strength of selection (RELAX) using the Datamonkey server [82].

## 3. Results

### 3.1. Topological Comparisons

Classical molecular evolution methods are based upon the establishment of homology (i.e., alignment) and tree building. Nonetheless, it can be informative to examine how alignment-free analyses compare to an alignment-based phylogeny [38]. In Figure 1, the *Alphapapillomavirus* alignment-based tree (Panel A) is largely recapitulated by the alignment-free trees based on trimer composition (Panel B), codon usage (Panel C), and base compositions (Panel D) where the low-risk 2 (LR2) viruses (green) did not cluster with the low-risk 1 (LR1) (blue) and high-risk (HR) virus types (Figure 1). Measures of topological congruence between trees counting the number of shared and unshared nodes with Matching Pair (MP) distances [83] demonstrated that the trees in Figure 1 are significantly more similar to each other (MP < 390) than to random trees (MP > 549, lower 95% CI = 559, *p* < 0.001). However, all three alignment-free trees placed the non-human primate viruses (*Alphapapillomavirus 12*, grey) within the LR2 clade instead of with the HR group, as found in the alignment-based tree. The discordance between alignment-based and alignment-free trees (Figure 1) could indicate driving features of additional information (e.g., non-coding elements) in the genomes that extend beyond aligned ORFs.

### 3.2. Sliding Window Analyses

To look for common patterns in unaligned genomes, we evaluated all *Alphapapillomavirus* types (n = 83) with sliding window analyses, finding the local number of CpG and APOBEC3 motifs in each 150 bp window while also measuring changes in GC content and local trimer composition; these analyses are mapped to nucleotide positions in the HPV16 reference genome (Figure 2A). For viral types infecting humans (Figure 2B), 70% of the windows with significantly higher concentrations of CpG sites corresponded to E4 (Figure 2B), which also had the highest GC content (green line in Figure 2A) and atypical trimer usage (blue line Figure 2A). For 39 of the *Alphapapillomavirus* types, CpG sites were nonrandomly distributed within E4 itself. A second region of high CpG concentration was found near the late polyadenylation region within the URR (Figure 2B). CpG sites in association with adenosine-rich motifs may be bound by antiviral proteins, such as zinc finger antiviral protein (ZAP) [84]; however, we did not find differences in observed vs. expected numbers of such motifs in genomes of alphapapillomaviruses (data not shown). Concentrations of APOBEC3 recognition sites accumulate linearly above expectations through the early ORFs, with exceptions for regions in E7 and E1 ORFs. The *Alphapapillomavirus* 12 viruses infecting non-human primates exhibit a pattern that is different from those infecting humans (grey plots Figure 2B).

### 3.3. Principal Component Analysis

To further examine the alignment-free clustering noted in Figure 1, we investigated *Alphapapillomavirus* genome content with Principal Components Analysis (PCA). Drift, as predicted by Brownian motion random-walk evolutionary models, results in non-recombining genomes within lineages to diffuse into overlapping regions if given enough time [85,86,87] (Figure 3 provides a conceptual model in relation to those expectations [85,86,87]). Thus, under drift, the ability to discriminate lineages diminishes with time (Figure 3), as indicated by recently diverged lineages being well-discriminated (see t = 1 in Figure 3) and more anciently diverged lineages/clades showing broad overlap (see t = 3 in Figure 3). In contrast, a common selection pressure should cause narrow convergence on a distinct optimum in the parameter space [88], and convincing evidence of this would be the convergence of lineages that are not the closest relatives of each other (see light blue and light green lineages at t = 3 in Figure 3B). Using unaligned *Alphapapillomavirus* reference genomes [5], reducing parameter space dimensionality with PCA among the 13 DNA sequence motifs (Appendix A) and simple base composition (A, C, G, T) showed a number of similarities (Figure 4A and Figure 4B, respectively). PCA incompletely segregated human high-risk HPVs from human LR1 virus types and incompletely segregated LR2 types from NHP-Alpha12 types (NHP), as previously illustrated by alignment-free clustering (Figure 1), and consistent with expectations for anciently diverged clades (see t = 3 in Figure 3A). Indeed, with the exception of *Alphapapillomavirus* 12 and LR2, GC content was significantly different between each clade (*p* < 0.001 in Mann–Whitney U and Bonferroni post hoc tests).

Tens of millions of years since the divergence of the LR1, LR2, and HR clades, clustering still remains that likely reflects their common ancestors. However, diffusion of the phylogenetic pattern is evident (Figure 4). Indeed, this is what is predicted with a Brownian motion drift process in which the displacement from an ancestral condition is the product of the parameter variance and time [89,90] (Figure 3). To evaluate patterns of recent type divergence, we leveraged 1278 complete *Alphapapillomavirus 9* HPV genomes (from GenBank, August 2022), including types, lineages, sublineages, and SNP variants. PCA of the 13 DNA sequence motifs and of nucleotide base compositions tended to isolate recently diverged types surprisingly well (Figure 5), as also reflected by individual base compositions (Appendix A). Based on the sequence motifs PCA, the strongest distinctions were for HPV16 (red) and HPV35 (grey), both from each other and from other types, whereas neither HPV52 and HPV67 nor HPV58 and HPV33 were fully discriminated (Figure 5A). Separation of types was more complete based on base composition where only HPV52 and HPV67 failed to discriminate (Figure 5B), and for which the first principal component largely reflected GC content (Appendix A). In addition to separating types, PCA of base compositions (Figure 5B) also discriminated among some variant lineages of HPV16, HPV58, HPV31, and HPV67. It should be noted that recently diverged lineages appear to have differentiated in ways that are type-specific (as in Figure 3A), without evidence of common ancestry or common adaptation (as expected in Figure 3B) to a similar ecological niche. In fact, HPV16, the most oncogenic HPV type, best reflects the ancestral nucleotide composition of *Alphapapillomavirus 9* (Appendix A).

The co-location of HPV52 and HPV67 in PCA space (Figure 5A,B) is not explained by recent common ancestry—they do not form a monophyletic group (see tree in Figure 5). An alternative explanation is that both contain features of their most recent common ancestor (denoted MRCA-x on Figure 5), whereas HPV58 and HPV33 have diverged (Figure 5B). A similar case can be made for the co-location of HPV58 and HPV33 genomes––that is, though their base compositions have diverged (Figure 5B), they still overlap in the PCA space based on more complex shared sequence motifs (Figure 5A) consistent with their recent common ancestor (denoted MRCA-y on Figure 5). The PCA results are consistent with Brownian motion [91] acting upon ancestral states of HPV isolates manifesting features of genetic drift [86,90,92].

### 3.4. Ancestral State Reconstruction

To understand how the DNA motifs in HPV genomes have changed over time, we utilized Brownian motion ancestral state reconstruction (see Figure 6, and Appendix A) [90,93]. Of the 13 DNA sequence motifs, 4 exhibited more ancestral branch change than expected (Figure 6): the number of APOBEC3 recognition sites (11 branches), the number of palindromic regions (8 branches), and the number of CpG sites and TLR9-stimulating motifs on the same branches given that the latter are CpG-rich. HPV16 has the least CpG sites and the most palindromic motifs represented by multiple instances of punctuated change since the *Alphapapillomavirus* MRCA. However, there does not appear to be *Alphapapillomavirus*-wide coordination of significant changes on ancestral lineages as would be expected from a selection pressure that is common to all of the alphapapillomaviruses. In a manner similar to the PCA of recently diverged HPV genomes (Figure 5), ancestral state reconstruction (Figure 6) suggests no obvious common evolutionary trajectory since the *Alphapapillomavirus* MRCA. For example, some lineages show marked decreases in CpG sites whereas others increase. Similarly, both increases and decreases in TLR9 sites are apparent in the ancestral lineages of LR1 and LR2. Thus, each clade seems to have its own specific history with respect to the evolution of these motifs.

As with PCA (above), to explore recent HPV oncogenic type variant divergences, we compared ancestral states for all *Alphapapillomavirus 9* variant reference genomes from PAVE, including HPV18 and HPV45, as an outgroup to root the *Alphapapillomavirus 9* tree (Figure 7, and Appendix A). Brownian motion reconstruction revealed that gradual changes for recently diverged variants have resulted in markedly different trajectories. For example, the number of inverted repeats with potential for secondary structure increases in HPV16/31 but was reduced among HPV52 variants, while the number of perfect palindromes increased in both HPV16 and HPV52 variant lineages (see Appendix A). HPV31 and HPV33 have numbers of palindromes that reflect the ancestral condition, implying convergent reduction in HPV67 and HPV58. The number of CpG sites is reduced in variants of HPV16, whereas CpG sites are more numerous in HPV18 and HPV45 than in any *Alphapapillomavirus 9* type. HPV31 variants, in particular, are closest to the predicted number of ancestral APOBEC3 sites, with increases in these sites for HPV35 and HPV33/58. In the outgroup, HPV45 has markedly fewer APOBEC3 sites in comparison to all other types examined.

These analyses did not suggest a common evolutionary mechanism that would explain how HPV16, HPV18, and HPV58 are among the most prevalent and/or most oncogenic of the HR-HPV viruses. Nevertheless, visual inspection of CpG and APOBEC3 content across *Alphapapillomavirus 9* variants and the outgroup lineages suggests an inverse relationship (Figure 7). Indeed, with a false-discovery rate of 5%, the extant values for CpG and APOBEC3 motifs on the coding strand of HPV16 and HPV18 variants proved to be inversely correlated: (R = −0.655, *p* = 0.006) (R = −0.955, *p* = 0.00006), respectively. However, this kind of pairwise correlation fails to account for phylogenetic relatedness and the non-independence of closely-related viruses (see also [29]).

### 3.5. Phylogenetic Independent Contrasts

The inverse pairwise correlation between CpG and APOBEC3 motifs (e.g., HPV18 variants in Figure 7) could result from the close phylogenetic relatedness of variants within lineages compared to the distance between lineages. Phylogenetic Independent Contrasts (PIC) provides a strategy to tease out these possibilities [86,90]. We looked for phylogenetic correlations among the 13 DNA motifs and base composition (A, C, G, T) in viral genomes at three hierarchical levels: all 83 *Alphapapillomavirus* reference types, 25 types in the human HR clade (*Alpha5/6/7/9/11*), and separately in the variants within *Alphapapillomavirus 7* and *Alphapapillomavirus 9*, as well as in the clade combining *Alphapapillomavirus 5* and *Alphapapillomavirus 6* in order to have similar degrees of freedom for variants.

Using the coding strand, base compositions showed correlations with PIC that are expected from Chargaff’s Second Parity Rule [94] (i.e., significant positive correlations for A with T, C with G, and negative for all other combinations), except that in *Alphapapillomavirus 9*, A and T did not exhibit the expected positive correlation (Appendix A). Additional significant correlations (FDR Q ≤ 0.01) using PIC on DNA sequence motifs are shown in Appendix A at each hierarchical level. Notably, the inverse correlation of CpG and APOBEC3 sites remains significant within the HR and *Alphapapillomavirus 7* clades (Appendix A). The number of palindromic regions (inverted repeats and perfect palindromes) was inversely related to CpG content across the *Alphapapillomavirus* and HR clades yet was not significant within any HR species subset. The only commonality among recently diverged HR species groups was that TLR9 and APOBEC3 motifs were significantly correlated in the opposite direction in *Alphapapillomavirus 7* and *Alphapapillomavirus 9* (Appendix A). Taken together, the results of PIC suggest that coordinated changes among the 13 motifs are not uniform through time, as would be expected if alphapapillomaviruses were adapting to the same niche or evolving toward a limited number of selective optimal genome features. This observation is also consistent with drift.

### 3.6. Codon-Based Selection

Lastly, we also evaluated the role of Darwinian (i.e., positive) selection using nonsynonymous to synonymous substitution rates (dN/dS). Examining the aligned non-overlapping ORFs resulted in 8676 aligned nucleotides (of which 6165 were variable), representing 2892 aligned amino acid positions (of which 2106 were variable). Using HyPhy [81] with a GTR model, the phylogeny- and genome-wide dN/dS was estimated to be 0.1743, indicating strong purifying selection. Clade-specific differences in the strength of selection were examined with RELAX [95]. The hypotheses with the highest likelihood ratios (LR) included a significant relaxed selection strength (K) in the LR2 clade (K = 0.63; LR = 50.72) and a significant increase in selection in the *Alphapapillomavirus 12* clade relative to all *Alphapapillomavirus* (K = 1.12; LR = 43.26) (Appendix A). In terms of selection on aligned sites instead of branches, when considering *Alphapapillomavirus* as a group, MEME [25] found purifying selection at 277 amino acid sites (*p* < 0.05) but no sites under diversifying selection after curation with G-blocks [96] (Appendix A).

## 4. Discussion

In this report, we focus on the role of non-coding DNA motifs and the evolution of alphapapillomaviruses. Rather than recovering patterns of nucleotide evolution that would suggest a common selective framework for alphapapillomaviruses within the cervicovaginal niche, we document non-coding changes occurring in a highly lineage-specific manner. These results are most consistent with directional mutation pressure and neutral evolution (i.e., genetic drift as described by Sueoka [97,98]). Given the rarity of positive Darwinian selection [26] and recombination, new HPV variants emerge containing different combinations of non-coding motifs as we describe from our analysis of 13 higher-order nucleotide motifs. Surprisingly, even nucleotide composition alone discriminated closely related types, e.g., alpha-9 types (compare Figure 3 and Figure 5). The evolution of HPV genomes represents independent yet successful genetic drift away from ancestral niche-adapted genotypes [99] escaping the accumulation of unfavorable mutations and the speed of Muller’s Ratchet [100].

In addition to alterations to the number of CpG, APOBEC3, and TLR9 sites, there have been significant changes to the number of palindromic regions, each of which can result from simple changes in overall nucleotide (A, C, G, T) composition [101,102]. A progressive loss of CpG sites was identified within the more prevalent HR-HPV types. This included a 30% reduction from the MRCA of all alphapapillomaviruses to the MRCA of HR-HPV, and, thereafter, an additional 12% decrease from the MRCA of HR-HPV to the MRCA of *Alphapapillomavirus* 9. Globally, HPV16 is the most prevalent type, and it notably possesses less than half the CpG sites inferred to have been present in the MRCA of all *Alphapapillomavirus* types. Undoubtedly the viable lower limit of CpG sites is constrained in part by the overlapping reading frames of E2 and E4, either of which may not be able to vary synonymously without altering the other reading frame non-synonymously (Figure 2). Intriguingly, a second region of increased CpG sites across *Alphapapillomavirus* corresponds to the 5’ URR containing a nuclear matrix association region [103,104,105] where viral genome-wide association studies (VWASs) consistently find SNPs associated with increased carcinogenicity [28,30,32,33].

The diverse family of apolipoprotein B editing and catalytic (APOBEC) enzymes play critical roles in host defense against a wide range of viruses through cytosine deamination of exposed ssDNA. APOBEC3 enzymes are also implicated in precancerous genome instability [51,106,107,108,109,110]. Like methylation, APOBEC3 activity is upregulated in HPV infections [55] and their signature mutation type [111] is associated with viral clearance [27] and the mutational footprint left on neoplastic host cells [48]. Whereas APOBEC3 sites are nonrandomly distributed across many human DNA viruses [27,50], we found more on the transcribed negative-strand than on the coding positive-strand of *Alphapapillomavirus* genomes. This could represent selection to minimize sites on the coding strand, given the tendency of APOBEC3 enzymes to also edit mRNA when they are over-expressed [112], which corresponds well to the inverse correlation of these motifs in Phylogenetic Independent Contrasts (Appendix A). Similarly, adenosine deaminases acting on RNA (ADAR), host defenses against viral RNA create a regime that would not favor use of adenosine on the coding strand, which could explain a coordinated paucity of antisense APOBEC3 (i.e., AGA/UGA in transcripts) motifs [113].

It was surprising that something as primordial as base composition recapitulated *Alphapapillomavirus* phylogeny and also strongly discriminated *Alphapapillomavirus 9* types by PCA. Clearly, there is more encoded in the genomes of alphapapillomaviruses beyond codon and amino acid usage, alternative splicing of mRNA transcripts [114], and the efficiency afforded by overlapping reading frames [115]. Even simple nucleotide motifs such as tandem repeats of guanine are prone to simultaneous oxidation by glutathione [64], the lack of which has been proposed as a risk factor for the development of cervical cancer [116]. In addition to a tight cytosine balance, nucleotide composition could be under selection on silent substitutions [117]––for example, through effects on the speed, efficiency, and accuracy of transcription and translation [29,118]. Discovering marked differences in the number of palindromic regions (perfect palindromes and inverted repeats) is interesting given that base composition biases lead to higher palindromic content when the bias favors complementary nucleotides (e.g., A and T) [119]. An increase in the number and diversity of locally arranged inverted repeats could be a source of variation in hairpins, cruciforms, and pseudoknots that are open to evolutionary tinkering given their functional significance for transcription, translation, and the influence of miRNA on HPV gene expression [120,121].

The predominance of purifying selection on amino acids is consistent with observations of other dsDNA viruses [122]. We did not find substantial evidence of diversifying selection operating on individual amino acid sites.

Our findings imply something quite different from the notion that *Alphapapillomavirus* diversity resulted from the onset of diversifying selection with ancestral occupation of a new ecological niche [123]. Such a release from stabilizing selection would be marked by evidence of positive selection [124], which is simply not observed. For example, consider the convergent adoption of nonsynonymous mutations (e.g., K417T, E484K, and N501Y) in the spike protein of multiple lineages of SARS-CoV-2 [125] that are under positive natural selection. The emergence and persistence of new variants based to a large extent on drift could result from genomic “robustness” to the mutation pressures imposed by innate intracellular host restriction factors, especially if accompanied by incomplete purifying selection [126]. Were this the case, one would expect the least mutationally compromised virus to be the most prevalent, and this seems to be the case when comparing HPV16 to other *Alphapapillomavirus* types (Appendix A). In conclusion, the evolutionary signature recovered in *Alphapapillomavirus* HPV genomes is one of marked purifying selection on amino acid changes (which bodes well for the effectiveness of HPV vaccines), and a lack of convergence towards any single common adaptive peak or strategy. Cytosine deamination patterns and strand-asymmetry are consistent with a directional mutation model of neutral molecular evolution and genetic drift [97,127].

## Figures and Tables

**Figure 1 viruses-15-01631-f001:**
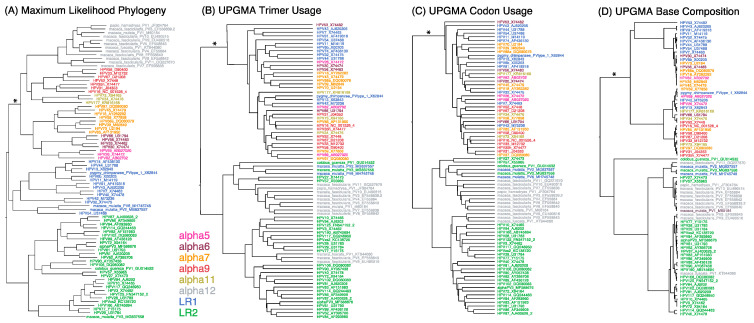
**Phylogenetic trees of alphapapillomaviruses based on different features.** Relatedness of each *Alphapapillomavirus* reference type was inferred from maximum likelihood (ML) phylogenetic analysis of aligned nucleotides in codons that are free to vary, excluding those in overlapping reading frames (**A**), as well as UPGMA analysis of unaligned genomes for trimer composition (**B**), codon usage (**C**), and base composition (**D**). Branch lengths are proportional to the amount of change. The internal branch uniting the high risk (HR) clade is denoted with an asterisk (*). HR-HPV types in the Alpha5/6/7/9/11 clade are colored pink, brown, orange, red, and gold, respectively; low-risk (LR) HPV type groups 1 and 2 are shown in blue and green, respectively; and non-human primate HPV types in Alpha12 are shown in grey.

**Figure 2 viruses-15-01631-f002:**
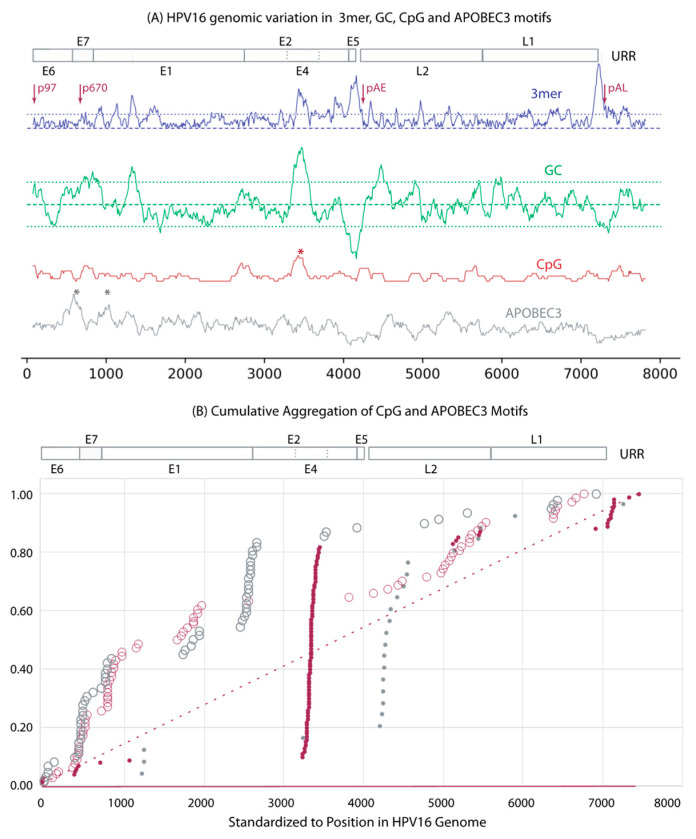
Sliding window analyses: (**A**) HPV16 is shown as a representative genome. The y-axes show the deviation from genome-wide expectations for trimer composition (3mer, purple), GC composition (GC, green), CpG number (CpG, red), and number of APOBEC3 motifs (APOBEC3, grey) across the genome, as determined by sliding window analyses; the average genome-wide expectation for each of these motifs is represented by the dashed line. The 95% confidence limits (dotted lines) are depicted for trimer composition and GC content. Positions with significant clustering of CpG or APOBEC3 sites are marked with asterisks. Promoter and polyadenylation sites are marked with arrows. (**B**) Cumulative analyses of all 83 *Alphapapillomavirus* reference types showing regions exhibiting significant clustering of CpG (solid circles) and APOBEC3 (open circles) sites. The genome position was standardized against the first codon of E6 in HPV16. Data from HPV *Alphapapillomavirus* types are in red. Data from non-human-primate-specific *Alphapapillomavirus* types are in grey. Red lines indicate expected cumulative distributions should sites be randomly distributed lacking clustering (solid, on the abscissa) and should sites exhibit random clustering (dashed).

**Figure 3 viruses-15-01631-f003:**
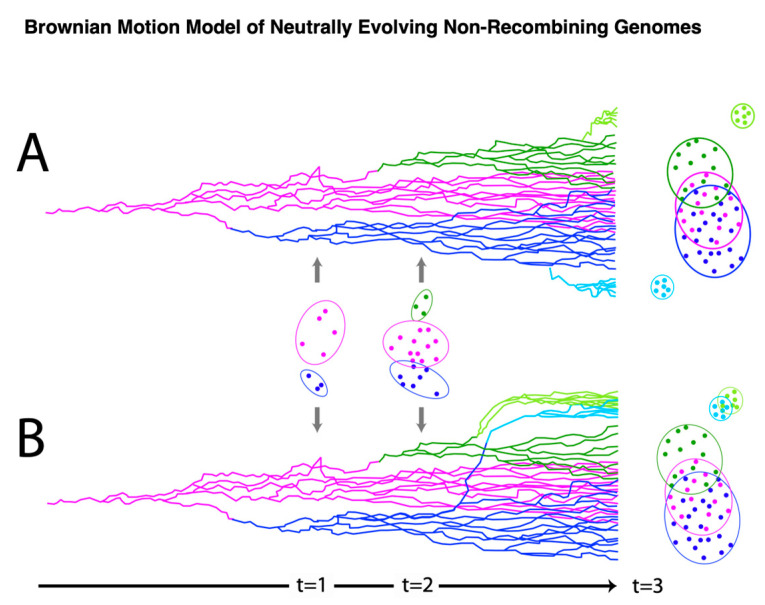
**Model of genetic drift.** Traits evolving under a Brownian motion drift model that readily distinguish recently diverged (asexual, nonrecombining) viral types (t = 1) are expected to broaden their distributions through random walk (t = 2), resulting in overlapping distributions (**A**) for more anciently diverged clades (t = 3). In contrast, the imposition of a common selective force will draw at least some unrelated lineages (**B**) to a more narrow, distinct, and coincident region of the parameter space (light blue and light green lineages).

**Figure 4 viruses-15-01631-f004:**
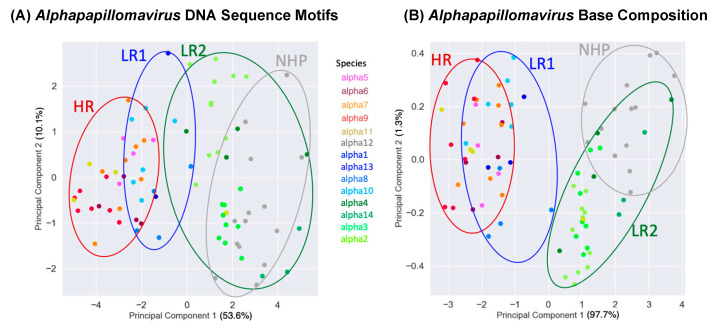
**Principle component analyses of HPV types.** Plot of second versus first principal components for 83 *Alphapapillomavirus* reference type genomes summarizing variation across 13 DNA sequence motifs (**A**), and summarizing variation for base compositions alone (**B**). Each point represents an *Alphapapillomavirus* type genome. Colors correspond to *Alphapapillomavirus* species, as in Figure 1. Ellipses circumscribe each of the two Low Risk (LR1 and LR2), High Risk (HR), and Non-Human-Primate-infecting (NHP) groups. Percentages of variation explained by each principal component are in brackets.

**Figure 5 viruses-15-01631-f005:**
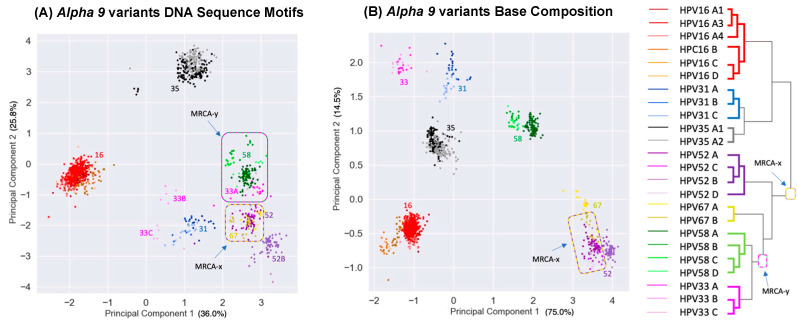
**Principle component analyses of HPV variants.** Plot of second versus first principal components for lineage variants of seven viral types in the *Alphapapillomavirus 9* species summarizing variations across 13 DNA sequence motifs (**A**), and summarizing variation for base compositions alone (**B**). Each point represents one of 1278 genomes. Colors correspond to HPV type and variant lineage, as indicated on the phylogenetic tree (at right). Percentages of variation explained by each principal component are in brackets.

**Figure 6 viruses-15-01631-f006:**
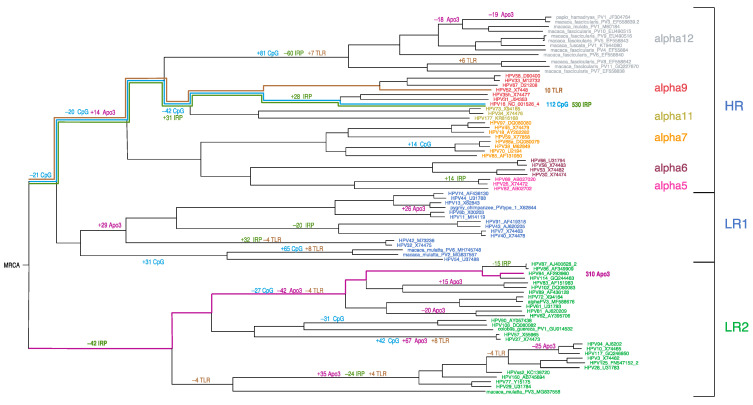
**Ancestral state reconstruction of alphapapillomaviruses.** Significant changes in ancestral states reconstructed on the ML tree for number of CpG sites (CpG), the number of APOBEC3 sites (Apo3), the number of Toll-like receptor 9 stimulating motifs (TLR), and the combined number of inverted repeats and perfect palindromes (IRP) was determined. Colored lines illustrate the ancestral paths from the most recent common ancestor (MRCA) of *Alphapapillomavirus* to the HPV type with the most extreme number of CpG sites (blue), the number of APOBEC3 sites (magenta), the number of Toll-like receptor 9 stimulating motifs (brown), and the combined number of inverted repeats and perfect palindromes (olive). Branch lengths are proportional to total genomic nucleotide change.

**Figure 7 viruses-15-01631-f007:**
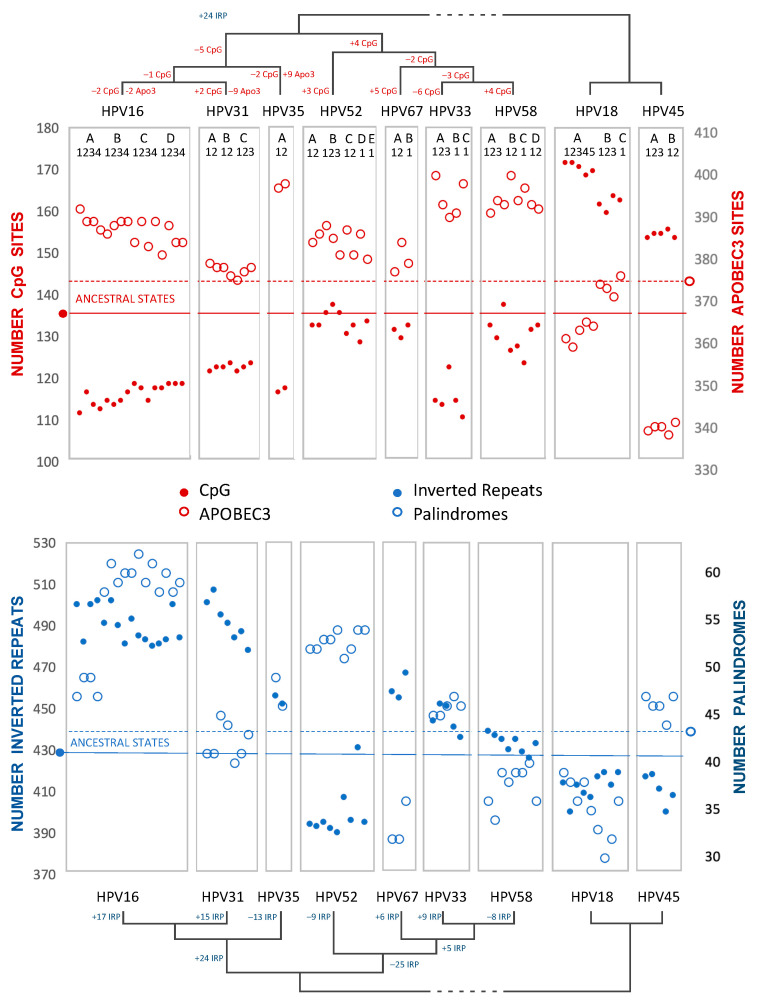
Ancestral state variation across HPV types, variants, and subvariants in the *Alphapapillomavirus 9* species. These panels contrast CpG sites with APOBEC3 sites (top), and inverted repeats with palindromes (bottom). The ancestral state inferred for each parameter is represented by a horizontal line. Each boxed group represents an HPV type in phylogenetic order with each point inside a box representing an HPV variant in alphabetical order (i.e., HPV16: A1/A2/A3/A4/B1/B2/B3/B4/C1/C2/C3/C4/D1/D2/D3/D4; HPV31: A1/A2/B1/B2/C1/C2/C3; HPV35: A1/A2; HPV52: A1/A2/B1/B2/B3/C1/C2/D1/E1; HPV67: A1/A2/B1; HPV33: A1/A2/A3/B1/C1; HPV58: A1/A2/A3/B1/B2/C1/D1/D2; HPV18: A1/A2/A3/A4/A5/B1/B2/B3/C; HPV45: A1/A2/A3/B1/B2).

## Data Availability

Alignments of nucleotide sequences, amino acid sequences and matrices of continuous variables with corresponding trees can be found at with accession number TB2:S30442at Treebase (www.trabase.org) as accessed on 1 April 2023.

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
