# Peer review of "Distinguishing Genetic Drift from Selection in Papillomavirus Evolution"

_viruses, 2023, doi:10.3390/v15081631_

Round 1

Reviewer 1 Report (Previous Reviewer 3)

I have read the resubmitted manuscript as well as the comments from previous reviewers. While I believe the manuscript has significantly improved after revision, there are some points that I authors did not address.

- The accumulation of CpG in E4 remains unexplained and not discussed. The authors proposed that overlapping ORFs, but its unclear why similar accumulation is not observed between E6-E7, L1-L2 and E8-E1. Is there a role for retaining CpGs in E4?

- Similarly, it's unclear if the unusual accumulation of CpGs in E4 is a mere consequence of higher GC-content in that region. The authors should normalize CpG count to the overall GC representation to that particular region. Is there information about RNA structural elements encoded in the regions that should be preserved?

- The question of selective pressure imposed by ZAP was not addressed. Indeed, many DNA viruses are sensitive to ZAP, including vaccinia virus, HCMV, etc. and while ZAP does not act on virus DNA, HPV mRNA would likely still be a substrate for this antiviral protein. Therefore, selective pressures imposed by ZAP would still be observable in HPV genomes. As previously suggested, the authors should evaluate if there is CpG depletion from A-rich regions, since those tend to be preferred targets of ZAP (PMID: 36075961).  

- I'm unaware of what the TLR9 evasion footprint looks like apart from depletion of CpG (PMID: 35311536). It's been shown that TLR9 has two binding sites: CpG and xCx (PMID: 29625894). Is there depletion of CpGs associated with local co-depletion of cytosines?

Author Response

"The accumulation of CpG in E4 remains unexplained and not discussed. The authors proposed that overlapping ORFs, but its unclear why similar accumulation is not observed between E6-E7, L1-L2 and E8-E1. Is there a role for retaining CpGs in E4?- Similarly, it's unclear if the unusual accumulation of CpGs in E4 is a mere consequence of higher GC-content in that region. The authors should normalize CpG count to the overall GC representation to that particular region. Is there information about RNA structural elements encoded in the regions that should be preserved?"

We thank the reviewer for the inquiry. In the paper, we merely noted that CpG sites are concentrated in the E4 region, "high concentrations of CpG sites corresponded to E4 (Fig. 2B) which also had the highest GC content." We did not suggest that the overlapping nature of ORFs or that the different GC content were responsible for the concentration of CpG sites in this region, though we did note the correspondence. Nor did we speculate as to why, for there are many possible reasons that we believe are outside of the scope of this paper including histone binding, intron retention, and simple constraints on overlapping reading frames.

As to the specifics of the reviewers' question, we also note that we did not ask in the paper whether the *number* of CpG sites in the genome was significantly different than what would be expected from the genomic GC content. We asked whether the *distribution* of CpG sites was non-random (i.e., concentrated). The reviewer's question regarding the *number* of CpG sites in E4, then, is a different one.

Nevertheless, regarding the reviewer's question, we can confirm that the *number* of CpG sites in the E4 region is not outside of what is expected from the GC content for E4 for any type.  However, the *distribution* of those CpG sites, even in the E4 region alone, remains non-random (concentrated in parts of E4) for 39 Alphapapillomavirus types. In the revised we have added this sentence stating, "For 39 of the Alphapapillomavirus types, CpG sites also were nonrandomly distributed with in E4 itself."

The reviewer's query about RNA structural elements will be addressed below.

"The question of selective pressure imposed by ZAP was not addressed. Indeed, many DNA viruses are sensitive to ZAP, including vaccinia virus, HCMV, etc. and while ZAP does not act on virus DNA, HPV mRNA would likely still be a substrate for this antiviral protein. Therefore, selective pressures imposed by ZAP would still be observable in HPV genomes. As previously suggested, the authors should evaluate if there is CpG depletion from A-rich regions, since those tend to be preferred targets of ZAP (PMID: 36075961). "

We apologize that we misunderstood the reviewer's point regarding ZAP. We had interpreted this only as a request to reference the ZAP literature.

In terms of analysis, finding that A rich regions are depauperate in CpG would be tautological.  That is, any 150 bp region that is entirely composed of A and T necessarily will have no CpG sites. It is also tautological that CpG content will be positively correlated with C and G content. Thus, what we believe the reviewer asks is whether the CpG content of a local region deviated from expectation in a way that is more negatively correlated with A than it is with T. From a statistical standpoint then, we have addressed the reviewer's question two ways: what proportion of C in a region is in a CpG site (CpG/C) and also what is the observed to expected ratio of CpG for that region given the G and C composition, and then are either of those more or less negatively correlated with A or with T content?

Fifty-one of 83 types show a negative local correlation between CpG/C and A content, of which 31 are significant. The average Spearman correlation coefficient, however, is very small (–0.04). On the other hand, all 83 types show a significant negative local correlation between CpG/C and T content, with an average correlation of –0.36; an order of magnitude stronger than that for A. Similarly, only 36 of 83 types have a significant correlation between observed-to-expected ratios of CpG versus local A content with an average correlation coefficient of –0.04. Meanwhile 79 of 83 types have a significant correlation between observed-to-expected ratios of CpG versus local T content with a similar average correlation coefficient of –0.04.

As such, we find no evidence in favor of the notion that A-rich ZAP sites are relevant to the distribution of CpG sites in alphapapillomaviruses and have removed the mention of it from the Discussion.

As it concerns such RNA-level matters (also raised above), we note that this paper has focused on DNA-level motifs.  That is not to say that RNA-level influences are not relevant to the heritability of DNA. However, the RNA-level matter is far more nebulous than the highly specifiable DNA motifs we measured. In part this is due to the plasticity of A:U/A:G bonding in RNA, but also relates to the plasticity of hnRNP sites and miRNA interactions. Similarly, we have not focused on transcription factor binding sites on DNA in this paper due to their well-known position-site weighting plasticity and given that there is evidence of papillomaviruses deviating strongly on those motifs from what is typical for the host DNA. We do not doubt that all of the foregoing are relevant and could be informative regarding the evolution of papillomaviruses. However, we are unconvinced that the binding site specificities are sufficiently well known for the wide array of hosts like cows, porpoises, bats, rabbits etc.

"I'm unaware of what the TLR9 evasion footprint looks like apart from depletion of CpG (PMID: 35311536). It's been shown that TLR9 has two binding sites: CpG and xCx (PMID: 29625894). Is there depletion of CpGs associated with local co-depletion of cytosines?"

We would direct the reviewer's attention to the references cited by us. In particular, King et al (1998; PMID: 9770537) which identifies a variety of TRL9 suppressing motifs some of which actually include CpG sites.  Thus, the matter is far more complex than the reviewer suggests. CpG sites are present both in TLR9 stimulatory and in TLR9 suppressing motifs. 

As to the latter question, yes of course there is local depletion of CpG where there is local depletion of C. But this is tautological as noted above. 

Reviewer 2 Report (Previous Reviewer 2)

The authors have addressed most of my comments appropriately, but some have yet to be corrected. HP16 should be corrected to HPV16 (line 482). Also, the newly added ORF map in Figure 2 seems to be incorrect; E6 (151 aa) should be larger than E7 (98 aa).

Author Response

We thank the reviewer for their attention to detail and have made both corrections.

Round 2

Reviewer 1 Report (Previous Reviewer 3)

The authors addressed all my concerns, however, I think the analysis of potential targets of ZAP (CpGs in an A-rich context) should be included, or at least discussed, in the manuscript. Based on the frequency of CpGs of alphapapillomaviruses presented in this study, it is very likely that virus RNA is targeted by this protein. If it doesn't, then the virus must encode an evasion mechanism. This observation made by the authors will substantiate future investigation of ZAP during HPV infection.

Author Response

This manuscript is a resubmission of an earlier submission. The following is a list of the peer review reports and author responses from that submission.

Round 1

Reviewer 1 Report

Burk and colleagues present an intriguing manuscript that explores the evolution of papillomaviruses using non-traditional approaches. However, as a reviewer, I find it challenging to grasp the specific focus of the authors' investigation and the resulting conclusions. It appears that the authors analyze a selected set of sequence features, but the rationale behind their selection and the interpretation of the figures remain unclear.

Before delving into the potential scientific implications of this work, I believe several issues need clarification. Firstly, additional details on the methods employed, including the provision of code for all calculations, would greatly enhance reproducibility. Secondly, I am curious about the procedure used to delete overlapping open reading frames (ORFs). Did this process consider codons due to the differing open reading frames? Additionally, were motifs that spanned deletions, thereby creating non-continuous sequences, also removed?

Furthermore, I would like to address concerns regarding Figure 1. While the authors claim that Figures 1B-D are similar to Figure 1A, I disagree. It would be beneficial to quantify the degree of similarity to support this assertion. Additionally, in Figure 2A, it is unclear what the Y-axis represents and why only HPV16 is examined.

Regarding Figure 2B, the uniform distribution of CpGs along the HPV16 genome raises questions about their accumulation solely in the E4 protein. I am curious about how the authors defined the E4 protein—did they rely on the splice acceptor or a random stop codon? Given that this analysis includes 85 genomes, it is important to consider potential errors associated with this observation.

In reference to Figure 3, I seek clarification on whether the presented data reflects empirical findings or if it is a model representation.

Considering the application of Brownian motion to the evolution of traits, I question its suitability for motifs, which are components of the genome. Can motifs be considered independent traits in this context?

Moving to Figure 4, the statement regarding the diffusion of the phylogenetic pattern is made without quantitative support. Furthermore, it is not evident that the depicted motifs are phylogenetic, as Figure 1 indicates a discordance between phylogeny and UPGMA.

In Figure 5, I am curious about the rationale behind selecting those two most recent common ancestors (MRCAs).

Lastly, Figure 6 highlights that four motifs display more ancestral change than expected. It would be valuable to understand the methodology used for this assessment and the basis for determining the expectation.

Addressing these concerns and clarifying these points would be required allowing for a more confident evaluation of its scientific implications.

Reviewer 2 Report

The manuscript by Burk et al. takes an alignment-free approach to elucidating the evolutionary principles that shape the Alphapillomavirus genome. Viral genome sequences covering 83 types of Alphapapillomavirus, lineages/sublineages, and SNP variants were systematically examined to figure out the number and location of non-coding DNA sequence motifs and subjected to principal component analysis (PCA). PCA using 13 DNA sequence motifs and base compositions yielded genetic clusters best explained by a mechanism, genetic drift. Overall, the analysis is comprehensive and adequately done, and provides interesting concepts about the evolution of the HPV genome. 

Major comments: 

In general, genetic drift is a mechanism of evolution that occurs when random fluctuations in the frequencies of different versions of a gene/allele in a population cause changes in their relative proportions over time. However, the authors seem to view the concept from a different perspective. At the single molecule level, genetic drift described by the authors, which is a stochastic change in the viral genome sequence like Brownian motion, is likely to occur, but for this change to contribute to the evolution of the viral genome, it must get fixed at a reasonable frequency in the viral population. In any case, population bottlenecks are important in understanding genetic drift as an evolutionary process, and it is recommended to discuss what bottlenecks (e.g., transmission, disease regression, latency) are thought to act on the HPV virus population for viral evolution. 

For this reviewer, it is not clear what is meant by "non-random mutation burden associated with genetic drift" (line 381) and needs to be rephrased or explained. 

Minor comments: 

  1. Line 119: The two smallest open reading frames are E7 and E4. 

  1. Line 405: APOBEC should be defined when it first appears. 

  1. Line 451: “HP16” to be “HPV16 

  1. References are not complete; there are no papers from ref. 75.

Reviewer 3 Report

Burk and colleagues used various analytical methods to examine if purifying selection on synonymous substitutions has played a role in the evolutionary history of alphapapillomaviruses. Their analyses revealed phylogenetic clusters consistent with genetic drift, and showed that different evolutionary drivers have shaped papillomavirus genome composition, particularly in the representation CpG and APOBEC3 motifs. The study is well performed and clearly written, but there are some point that require further clarification.

Major Points:

1. The authors frequently use the term "non-coding DNA elements" to describe the type of nucleotide substitutions they studied when analysing alphapapillomavirus genomes. What I believe its meant is synonymous mutations in DNA sequences. The term non-coding DNA is frequently applied to regions of DNA that do not encode protein, such as promoter regions or untranslated regions. The use of this term is confusing and the authors should consider clarifying it or use a different term.

2. In Figure 2A, it would be useful to align open reading frames of HPV16 with the nucleotide position written in the x axis. This will make easier to visualize synonymous mutations in the context of the genome. 

3. The authors say in Figure legend 2 "Positions with significant clustering of CpG or APOBEC3 sites are marked with asterisks." Are there asterisks on the figure? I wasn't able to find them.

4. In Figure 4B, looks like 97% variation is explained by Base Composition alone. Are there significant variation in GC richness in HR/LR1 versus LR2/NHP? What are the selective pressures of this? The authors should plot GC content for each virus.

5. There seems to be a bias towards cytosine suppression. (Sup Figure 9) Both cytosine deamination aided by the enzymatic activity of APOBEC (PMID: 35597990) and the spontaneous deamination of 5-methylcytosine produced by DNA methyltransferases (PMID: 24020411) can lead to cytosine suppression. Can the authors quantify what percentage of synonymous mutations were due to the activity of APOBEC versus spontaneous deamination?

6. The zinc finger antiviral protein (ZAP) binds to CpG dinucleotides in virus RNA and targets such RNA for degradation. It has been shown that CpGs located in an A-rich context are subjected to the activity of ZAP (PMID: 36075961). The authors should calculate if A-rich genes were purged from CpG dinucleotides as a consequence of ZAP.

7. The calculation of CpG frequencies along the genome can be misleading, since GC-rich regions have higher probability of containing CpG dinucleotides than AT-rich regions. The authors should include observed/expected ratios of CpG for each individual genes on HPV.

Minor points:

Line 54 - Please define UPGMA

Line 63 - Please define SNP

Line 68 - Please define PV

Line 188 - Please define URR

Line 272 - What are higher-order sequence motifs? What motifs are over-represented in 58 and 33?

Line 417 - What is meant by "APOBEC3 sites also are open to attack during repair of 417 deaminated CpG sites"